# Pre-Procedural Lumbar Neuraxial Ultrasound—A Systematic Review of Randomized Controlled Trials and Meta-Analysis

**DOI:** 10.3390/healthcare9040479

**Published:** 2021-04-17

**Authors:** Tatiana Sidiropoulou, Kalliopi Christodoulaki, Charalampos Siristatidis

**Affiliations:** 1Second Department of Anesthesiology, School of Medicine, National and Kapodistrian University of Athens, Attikon University Hospital, Rimini 1, 12462 Athens, Greece; tsidirop@med.uoa.gr (T.S.); kalchris@hotmail.gr (K.C.); 2Assisted Reproduction Unit, Second Department of Obstetrics and Gynecology, Aretaieion Hospital, Medical School, National and Kapodistrian University of Athens, Vas. Sofias 76, 11528 Athens, Greece

**Keywords:** ultrasound, neuraxial, epidural, spinal, anesthesia, analgesia, space, lumbar

## Abstract

A pre-procedural ultrasound of the lumbar spine is frequently used to facilitate neuraxial procedures. The aim of this review is to examine the evidence sustaining the utilization of pre-procedural neuraxial ultrasound compared to conventional methods. We perform a systematic review of randomized controlled trials with meta-analyses. We search the electronic databases Medline, Cochrane Central, Science Direct and Scopus up to 1 June 2019. We include trials comparing a pre-procedural lumbar spine ultrasound to a non-ultrasound-assisted method. The primary endpoints are technical failure rate, first-attempt success rate, number of needle redirections and procedure time. We retrieve 32 trials (3439 patients) comparing pre-procedural lumbar ultrasounds to palpations for neuraxial procedures in various clinical settings. Pre-procedural ultrasounds decrease the overall risk of technical failure (Risk Ratio (RR) 0.69 (99% CI, 0.43 to 1.10), *p* = 0.04) but not in obese and difficult spinal patients (RR 0.53, *p* = 0.06) and increase the first-attempt success rate (RR 1.5 (99% CI, 1.22 to 1.86), *p* < 0.0001, NNT = 5). In difficult spines and obese patients, the RR is 1.84 (99% CI, 1.44 to 2.3; *p* < 0.0001, NNT = 3). The number of needle redirections is lower with pre-procedural ultrasounds (SMD = −0.55 (99% CI, −0.81 to −0.29), *p* < 0.0001), as is the case in difficult spines and obese patients (SMD = −0.85 (99% CI, −1.08 to −0.61), *p* < 0.0001). No differences are observed in procedural times. Ιn conclusion, a pre-procedural ultrasound provides significant benefit in terms of technical failure, number of needle redirections and first attempt-success rate. Τhe effect of pre-procedural ultrasound scanning of the lumbar spine is more significant in a subgroup analysis of difficult spines and obese patients.

## 1. Introduction

Neuraxial procedures are traditionally blind techniques where space is identified by palpation of anatomic landmarks and loss of resistance to air or saline is used to identify the epidural space [1]. Palpation is often inaccurate, depending on the experience of the anesthesiologist and objective difficulties related to obesity and spinal anatomy [2]; therefore, failed blocks and complications are common. The use of anatomical landmarks is responsible for incorrect calculation of a predetermined intervertebral space and cephalad misinterpretation of the desired level [3,4,5]. Repeated attempts and redirections of the needle may lead to accidental dural puncture during an epidural technique or traumatic/bloody taps during spinal anesthesia or lumbar puncture, adding discomfort to the patient and delaying treatment.

A plethora of studies is currently available describing the use of “pre-procedural” neuraxial ultrasound (also termed ultrasound-assisted) to identify relevant anatomy, determine the desired intervertebral space and give an accurate estimation of the preferred insertion site of the spinal or epidural needle. Despite compelling advantages that result from its use in terms of technical failure [6,7], there are still questions left unresolved pertaining to specific populations in which the major clinical benefit could be observed or prolonged procedural time might outweigh the alleged clinical benefits [8,9].

The aim of this review is to examine current evidence on the utilization of a pre-procedural neuraxial ultrasound and investigate specific predetermined outcomes in patients requiring neuraxial procedures, as well as in a subgroup of these with non-palpable landmarks due to obesity or difficult spinal anatomy.

## 2. Materials and Methods

### 2.1. Study Registration and Reporting

The presented review has been registered with International prospective register of systematic reviews (PROSPERO) (Registration No: CRD42019121278). The reporting followed the “Preferred Reporting Items for Systematic Reviews and Meta-Analyses (PRISMA) statement” [10].

### 2.2. Literature Search and Inclusion Criteria

The electronic databases MEDLINE, the Cochrane Central Register of Controlled Clinical Trials, Science Direct and Scopus were searched. The following keyword algorithm was used: (“Ultrasound” OR “ultrasonography”) AND (“epidural” OR “peridural” OR “subarachnoid” OR “spinal” OR “lumbar” OR “neuraxial”) AND (“analgesia” OR “anesthesia” OR “space” OR “puncture”). After the initial deduplication process, we conducted an initial title-abstract screening, excluding only totally irrelevant randomized controlled trials (RCTs). During the second stage of full-text assessment, only studies fulfilling our predefined criteria were considered eligible for our review. The research was limited by language (English only) and publication date (from 1 January 1980 to 1 June 2019). Our full search strategy is available in Appendix A. Eligible articles were also identified through searching the references of included studies, thus implementing a snowball procedure.

### 2.3. Eligibility Criteria

We sought full-text reports of RCTs with a parallel-group design (pre-procedural ultrasound vs non-ultrasound-assisted control groups). We predefined the “population” as adult patients (≥18 years old) undergoing lumbar neuraxial procedures; “intervention” as pre-procedural lumbar neuraxial ultrasound; “comparator” as a non-ultrasound assisted control group; and “primary outcomes” as:Technical failure rate defined as the need to (i) use alternative techniques (e.g., use of ultrasound in the conventional landmark group or vice versa) to achieve the effect or (ii) conversion to general anesthesia or (iii) discontinuation of the technique. First attempt success rate was defined as a single needle puncture with or without redirections of the needle to obtain the desired outcome (visualization of cerebrospinal fluid (CSF), injection of local anesthetic, threading of the epidural catheter).Number of needle redirections during the neuraxial procedure. We noted some discrepancies across studies (e.g., referred to as number of attempts (including number of redirections and new needle insertions)). We decided to include in the meta-analysis studies that reported number of redirections, as well as studies that included sum of number of redirections and new needle insertions, because every new puncture accounts for a new needle redirection in clinical practice.Total procedural time, defined as time to identify landmarks either by pre-procedural ultrasound or by conventional landmark palpation plus needling time (see below).Needling time (time from when needle touched the skin until desired outcome, e.g., visualization of CSF, injection of local anesthetic, threading of the epidural catheter).

### 2.4. Data Collection Process

Two authors (T.S. and K.C.) independently extracted data from original report, and a third author (C.S.) resolved discrepancies. Extracted data were transferred into an Excel spreadsheet that was designed for the purpose of this analysis. We planned to contact authors of original reports if additional unpublished data were required for analysis.

We collected information on year of publication, first author’s name, country of origin, type of procedure (epidural, combined epidural-spinal anesthesia (CSE), spinal or lumbar puncture), study sample size, patient characteristics (physical status, mean age, body mass index and sex), comparison groups, type of population/surgery/setting (e.g., obstetric patients, orthopedic surgery, emergency department), primary and secondary outcomes and main findings of the study.

Relevant data were extracted from the text, tables or graphs from each source study. If a study reported the median [interquartile range], this was converted to the mean and standard deviation as follows: the mean was estimated according to the method of Luo and colleagues [11] and the standard deviation was calculated according to the method of Wan and colleagues [12]. Finally, we rated the quality of evidence for each outcome, following the Grades of Recommendation, Assessment, Development and Evaluation (GRADE) Working Group system [13].

### 2.5. Risk of Bias Assessment

Limitations related to trial design and implementation were assessed with the Cochrane risk of bias assessment tool [14] and the Jadad Scale [15]. The Jadad Scale (or the Oxford quality scoring system) is a five-point questionnaire to independently assess the methodological quality of a clinical trial (randomization: 0–2 points; blinding: 0–2 points; description of dropouts: 0–1 point). Two authors (T.S. and K.C.) separately screened, reviewed and rated the items for each trial. We assessed publication bias using three methods: funnel plot with trim-and-fill analysis [16], Begg and Mazumdar rank correlation test [17] and Egger regression test [18], if at least five trials with at least one event each could be included. Publication bias analysis was performed using Meta-Essentials (Erasmus Institute, The Netherlands) [19].

### 2.6. Statistical Analysis

Review Manager software (RevMan version 5.3.5; Copenhagen, The Nordic Cochrane Centre, The Cochrane Collaboration 2014) was used to perform meta-analyses. The software estimates the weighted mean differences for continuous data and risk ratios for categorical data between groups, producing an overall estimate of the pooled effect. We expected most datasets to be heterogeneous, therefore, they were analyzed using a random-effects model, and are presented as the standardized mean difference (SMD) or relative risk (RR) with 99% CI. We conducted a meta-analysis if more than five trials reported similar outcomes. Further analysis was performed in obese and difficult spinal subgroups if the outcomes were reported in at least five studies. Study heterogeneity and inconsistency were evaluated using the Q statistic and I^2^, respectively. Significant heterogeneity was defined as *p* < 0.1 and inconsistency as I^2^ > 75%. Data from each trial were considered as per the intention-to-treat principle. We also computed numbers needed to treat (NNT) with 99% CI.

## 3. Results

This section may be divided by subheadings. It should provide a concise and precise description of the experimental results and their interpretation, as well as the experimental conclusions that can be drawn.

### 3.1. Characteristics of the Included Studies

A total of 4203 records were identified through the initial search. A detailed description of the selection process flow is provided in Figure 1. For eligibility, 269 full-text articles were assessed. Of these, only 32 RCTs met the inclusion criteria (Table 1 and Appendix A). The 32 studies involved 3439 adult patients undergoing neuraxial procedures. The average sample size was 107 subjects per study. Four articles involved 384 patients in the emergency room (ER) undergoing a lumbar puncture [20,21,22,23], while the remaining 28 articles involved 3031 patients undergoing neuraxial anesthesia. Nine studies examined 1169 patients undergoing epidural anesthesia [24,25,26,27,28,29,30,31,32], six studies examined 593 patients undergoing CSE anesthesia [33,34,35,36,37,38], and 13 studies examined 1293 patients who underwent spinal anesthesia [39,40,41,42,43,44,45,46,47,48,49,50,51]. Notably, 22 studies concerned obstetric patients. Eleven RCTs involved patients with expected technical difficulties, due to obesity [22,26,38,41,42,45,47,51] difficult spines due to scoliosis, kyphosis or previous spinal surgery [26] or due to impalpable spinous processes, interspinous spaces or vertebral columns [23,26,41,42,44]. In four of the abovementioned studies [21,22,23,47], subgroup analysis was performed, while the remaining seven studies were designed to include only patients for whom difficulty was expected.

All trials except from three compared the pre-procedural neuraxial ultrasound with the conventional palpation method for the identification of landmarks. The study by Grau et al. [35] involved three groups; the third group compared real-time epidural identification with the aforementioned other two groups: patients of that group were not included in our meta-analysis. The study by Kawaguchi et al. [28] compared the conventional median epidural technique by palpation to the pre-procedural ultrasound-guided lumbar technique to achieve ipsilateral-dominant epidural block. The study by Wilkes et al. [32] compared pre-procedural ultrasound neuraxial identification of landmarks with a group where a sham ultrasound was performed before palpation, without skin marking; this was done for blinding reasons.

In the vast majority of the studies, two ultrasound views were used for landmark identification: the oblique paramedian sagittal view to identify the sacrum and lumbar laminae and count intervertebral levels, and the transverse median view to investigate the posterior complex (ligamentum flavum, epidural space and posterior dura) and anterior complex (anterior dura, posterior longitudinal ligament and posterior vertebral body), assess the size of the interlaminar acoustic window and thus identify the best puncture site. In the control group, identification of landmarks was done by palpation of the intercristal line and the intervertebral spaces. Choice of the best puncture site was left at the discretion of the physician performing the procedure.

Six trials were conducted in the United States, four each in Germany and Turkey, three each in Egypt, Ireland and Canada, two in China, and one each in Australia, the United Arab Emirates, India, Singapore, Iran, Japan and Italy.

### 3.2. Risk of Bias Assessment

The risk of bias assessment showed the RCTs to be of moderate quality, with the exception of the blinding of patient and study personnel, which was of very low quality (Figure 2 and Appendix A).

### 3.3. Failure Rate

All of the RCTs reported either a failure rate or did not report any case of technical failure, as described in our study protocol. For all patients, the combined RR was 0.69 (99% CI, 0.43 to 1.10), Z = 2.06, *p* = 0.04, p for heterogeneity = 0.38, I^2^ = 6%, Q = 23.33, and the NNT was 47 (99% CI, 25.7 to 259.1) (Figure 3A). For obese and difficult spine patients (11 studies; 625 patients), the RR was 0.53 (99% CI 0.22–1.27), Z = 1.86, *p* = 0.06, p for heterogeneity = 0.45, I^2^ = 0%, Q = 8.55, and the NNT was 15 (99% CI, 8.4 to 55.8) (Figure 3B). Publication bias was checked with the funnel plot that showed some asymmetry (Appendix A), however, with a minor effect on the effect estimate, as indicated by the trim and fill analysis (estimated RR = −0.29 (99% CI, −0.69 to 0.11)). Neither the Begg and Mazumdar rank correlation test (*p* = 0.552) nor Egger’s regression asymmetry test (*p* = 0.298) confirmed the presence of significant publication bias. Similar results concerning publication bias were obtained in the difficult spine/obese patients’ subgroup analysis (Appendix A).

### 3.4. First-Attempt Success Rate

Nineteen RCTs reported the first-attempt success rate in the text, tables and figures in 1897 patients and were analyzed. The combined RR was 1.5 (99% CI, 1.22 to 1.86), Z = 4.96, *p* < 0.0001, *p* for heterogeneity <0.0001, I^2^ = 78%, Q= 75.31, and the NNT was 5 (99% CI, 3.4 to 5.6) (Figure 4A). Five RCTs reported data on 374 difficult spines and obese patients with an RR of 1.84 (99% CI, 1.44 to 2.34), Z = 6.49, *p* < 0.0001, p for heterogeneity = 0.72, I^2^ = 0%, Q = 2.07, and the NNT was 3 (99% CI, 2.1 to 4) (Figure 4B). The funnel plot of all trials showed some asymmetry (Appendix A), however, with a minor effect on the effect estimate (estimated RR = 0.41 (99% CI, 0.14–0.68)). Neither the Begg and Mazumdar correlation test (*p* = 0.221) nor Egger’s regression asymmetry test (*p* = 0.374) confirmed the presence of significant publication bias. Results concerning publication bias in the difficult spine/obese patients’ subgroup analysis are reported in Appendix A.

### 3.5. Number of Needle Redirections

Twenty-six RCTs reported the number of needle redirections, examining a total of 2822 patients. The SMD was −0.55 (99% CI, −0.81 to −0.29), Z = 5.44, *p* < 0.0001, *p* for heterogeneity <0.0001, I^2^ = 84% and Q = 160.4 (Figure 5A). Nine RCTs reported data on 593 difficult spines and obese patients (Figure 5B). Pre-procedural ultrasound reduced the number of needle redirections with an SMD of −0.85 (99% CI −1.08 to −0.61), Z = 9.26, *p* < 0.0001, p for heterogeneity = 0.36, I^2^ = 9%, Q = 8.98.

Publication bias was tested and the funnel plot showed asymmetry (estimated Hedges’ *g* = 0.55 [99% CI, 0.28 to 0.83], number of missing studies = 0) (Appendix A). However, neither the Begg and Mazumdar rank correlation test (*p* = 0.494) nor Egger’s regression asymmetry test (*p* = 0.515) confirmed the presence of significant publication bias. Results regarding publication bias in the difficult spine/obese patients’ subgroup analysis are reported in Appendix A.

### 3.6. Procedure Time

Total procedure time (time to identify landmarks and needling time) was reported in 11 RCTs. The SMD was 0.80 (99% CI, −0.38 to 1.97), Z=1.75, *p* = 0.08, p for heterogeneity < 0.0001, I^2^ = 97%, Q = 375.25 (Appendix A). Seven RCTs reported data on difficult spines and obese patients (Appendix A). The SMD for these seven trials was 0.23 (99% CI −1.10 to 1.56), Z = 0.45, *p* = 0.66, p for heterogeneity <0.0001, I^2^ = 96% and Q = 144.96.

Needling time (time from when needle touched the skin until the desired outcome was obtained) was reported in 16 trials with an SMD of −0.33 (99% CI, −0.74 to 0.09), Z = 2.02, *p* = 0.04, p for heterogeneity <0.0001, I^2^ = 89% and Q = 144.30 (Appendix A), while data from seven RCTs involving difficult spines and obese patients only had an SMD of −0.23 (99% CI, −0.85 to 0.39), Z = 0.96, *p* = 0.34, p for heterogeneity <0.0001, I^2^ = 84% and Q = 37.95 (Appendix A).

### 3.7. Quality of Evidence

We summarized the quality of evidence for the aforementioned outcomes based on the GRADE Working Group system for the whole population and for the subgroup involving difficult spines and obese patients. Based on our findings, the quality of evidence concerning the technical failure rate was high for the whole population under study and the subgroup of obese and difficult spinal patients. The quality of evidence for the other outcomes ranged from low to very low (Appendix A).

## 4. Discussion

This systematic review and meta-analysis evaluates the effect of a pre-procedural lumbar ultrasound on neuraxial procedures. Based on 32 RCTs and 3439 patients, our results show that pre-procedural ultrasound reduces the failure rate and number of needle redirections, while it increases the first attempt success rate. It does not prolong overall procedure time and might shorten needling time, although the clinical significance of such finding remains uncertain. The quality of evidence for technical failure is high because of the large number of trials involved and the low heterogeneity observed for this outcome. For the remaining outcomes, the quality of evidence ranges from very low to low because of the inconsistency and heterogeneity in absolute effects observed for all outcomes, as well as the high heterogeneity of the included studies. We have also sought to observe the effect of pre-procedural ultrasound on a predetermined subgroup, involving difficult spines and obese patients. In these subgroups, we observed similar differences in outcomes with the main meta-analysis, along with low heterogeneity concerning the first three outcomes, resulting in an overall higher quality of evidence. We are, therefore, more confident to recommend its use in these populations.

Several quantitative reviews [8,9,52,53] and two quantitative analyses [6,7] have questioned the effect of a pre-procedural ultrasound for reducing the incidence of failure in neuraxial procedures. Similar to our review, the trials were heterogeneous but their conclusions were in line with ours. The review by Shaikh et al. [7] summarized results from trials including both real-time and pre-procedural ultrasound, and pediatric and adult patients. They reported a 79% reduction in the risk of failure of spinal or epidural anesthesia, while Perlas et al. [6] evidenced a 49% risk reduction. We found a 31% reduction in the general cohort and a 47% reduction in the two subgroups, of difficult spines and obese patients. Thus, we recommend a pre-procedural ultrasound to avoid serious complications associated with increased attempts and traumatic techniques used to identify the epidural or subarachnoid space, such as spinal hematoma, traumatic cord injury or post-dural puncture headache.

We found that the pre-procedural ultrasound reduced the number of needle redirections. Shaikh et al. [7] found a significant decrease in the number of needle redirections (MD= −1.00, *p* < 0.001) and Perlas et al. [6] reported an MD of −0.75 (*p* < 0.001). In our meta-analysis, the MD was comparable to these reports (MD = −0.78; 99% CI −1.11 to −0.44, *p* < 0.0001).

In addition, the first-attempt success rate was significantly increased by a pre-procedural ultrasound. One in five patients in any clinical setting and one in three patients with difficult or impalpable landmarks due to obesity or other factors would benefit from its use. The main risk of increased attempts and traumatic techniques to identify the epidural or subarachnoid space are complications, such as spinal hematoma or traumatic cord injury [54,55,56,57]. These are auspiciously rare events; none of the studies included reported such outcomes. Information on more frequent complications, such as post-dural puncture headaches or backache was limited and inconsistent among studies, so that the results could not be synthesized. A firm conclusion on the incidence of minor and major complications with or without pre-procedural ultrasound is not possible and assumptions can be based on the observed reduction in the number of needle redirections and increased first-attempt success rate.

An important point for the expansion of pre-procedural ultrasound is that its use should not be limited to experts’ hands. In most published trials, operators were proficient in spinal ultrasound; a small number of trials involved residents or fellows with no previous experience [21,24,29,42] or various degrees of ultrasound experience [23]. More studies designed to address operator experience in ultrasound handling are warranted, in order to determine the prerequisites of adequate spinal ultrasonography. Moreover, the image resolution of modern US machines is constantly improving, therefore, imaging accuracy is expected to encourage its widespread use. Interestingly, the authors of the earlier published studies [26,27,34,35], utilizing ultrasound equipment with modest imaging, also found significant differences in terms of the number of needle redirections and first-attempt success rate.

Several limitations in this systematic review need to be discussed. The included studies involved miscellaneous procedures and patients (e.g., obstetric, emergency room, surgical patients). We deliberately included the above-mentioned procedures to broaden the sample of the analysis and increase the power of synthesis. It is likely that this contributed to the observed heterogeneity.

The most serious problem evidenced by this review was the lack of blinding among trials. Some studies attempted to circumvent this problem. Three studies [29,32,33] performed an ultrasound of the lumbar spine in both study groups to blind patients on group allocation. In the sham-US group, no skin marking was performed. Nomura et al. [22] carried out skin markings for each group with different shapes, using an ultraviolet ink pen not visible without an ultraviolet light source, before randomization. When the patient was randomized, then the investigator marked the desired puncture site with a visible pen according to the allocated group. Therefore, the operator of the lumbar puncture and the patient were both blinded to group allocation. While blinding the operator performing the ultrasound is often impossible, there is no reason to avoid blinding the personnel and patients by means of a sham ultrasound in the control group.

Other potential sources of biases are related to the study design (e.g., pooling together midline and paramedian approaches). Although the majority of studies used a midline transverse approach to the lumbar spine, two studies [48,49] used a paramedian approach in the experimental group to perform spinal anesthesia. In two studies [29,49], a predetermined interspace level (L5–S1) was used for the experimental group compared to the best interspace level palpated for the control group. We strongly believe that this negates one of the major benefits of pre-procedural ultrasound, which is the choice of the wider and better accessible intervertebral level insonated.

Our review has several strengths, such as: a comprehensive search strategy that included four major databases; a set of clinically meaningful predefined outcomes and subgroup analyses; a statistically “conservative” approach, implementing a random-effects model throughout to allow for treatment effects to differ across settings; a 99% CI to consider multiple primary outcomes’ testing. The latter was implemented to improve the effect estimation with respect to a 95% CI. The results were of high quality in the main outcomes, while the heterogeneity of studies in the subgroup analyses was low.

## 5. Conclusions

In conclusion, in our review, we examined a large number of patients undergoing lumbar neuraxial procedures in different clinical settings; the quantitative synthesis of the analysis evidenced a significant benefit of a pre-procedural ultrasound, in terms of technical failure, number of needle redirections and first attempt-success rate. The degree of the results’ heterogeneity suggests that these findings should be interpreted with caution. Importantly, the effect of pre-procedural ultrasound scanning of the lumbar spine was more significant in a subgroup analysis of difficult spines and obese patients. Based on the current results, future high-quality studies are needed to address and prove or disprove our findings.

## Figures and Tables

**Figure 1 healthcare-09-00479-f001:**
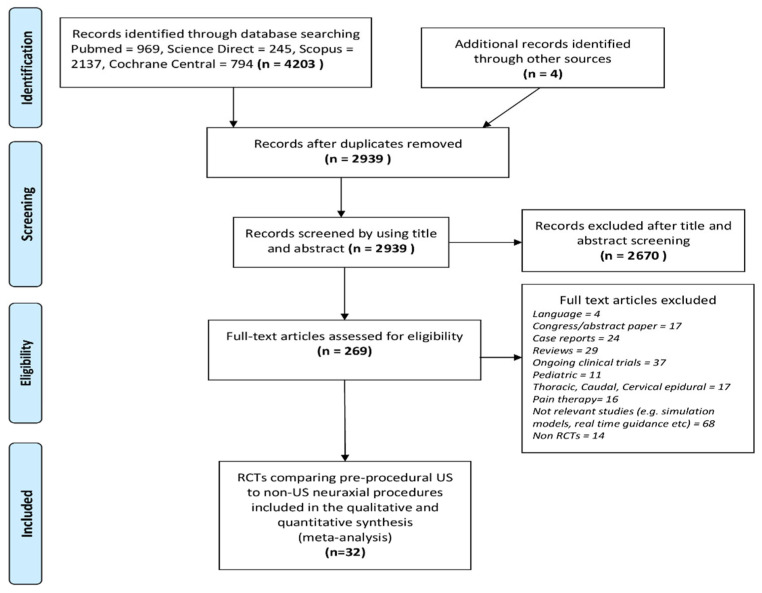
PRISMA flow diagram of this systematic review.

**Figure 2 healthcare-09-00479-f002:**
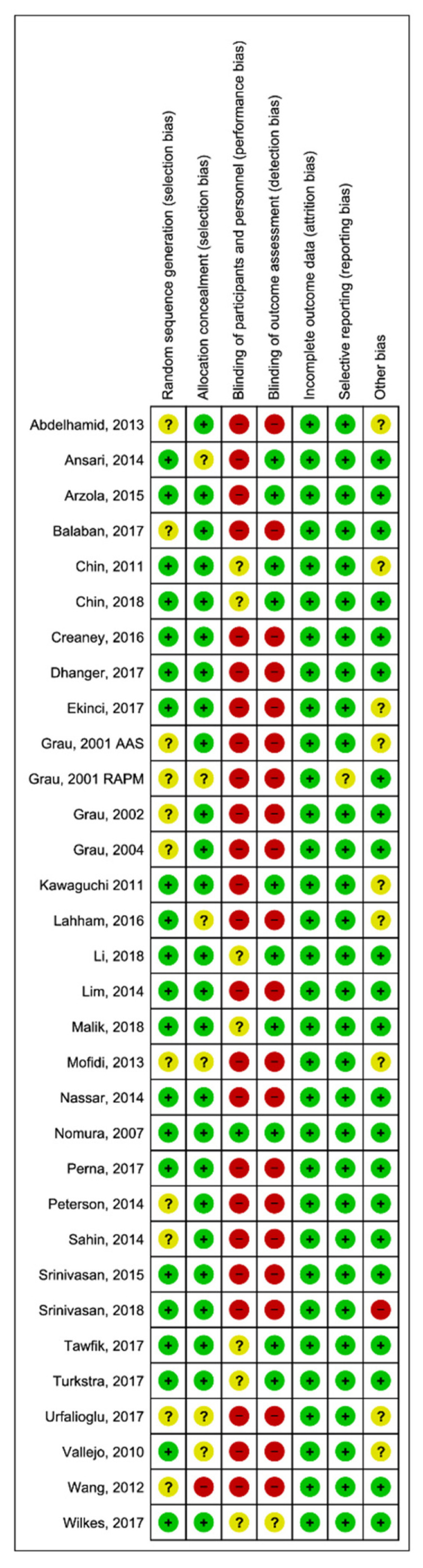
Risk of bias of included studies.

**Figure 3 healthcare-09-00479-f003:**
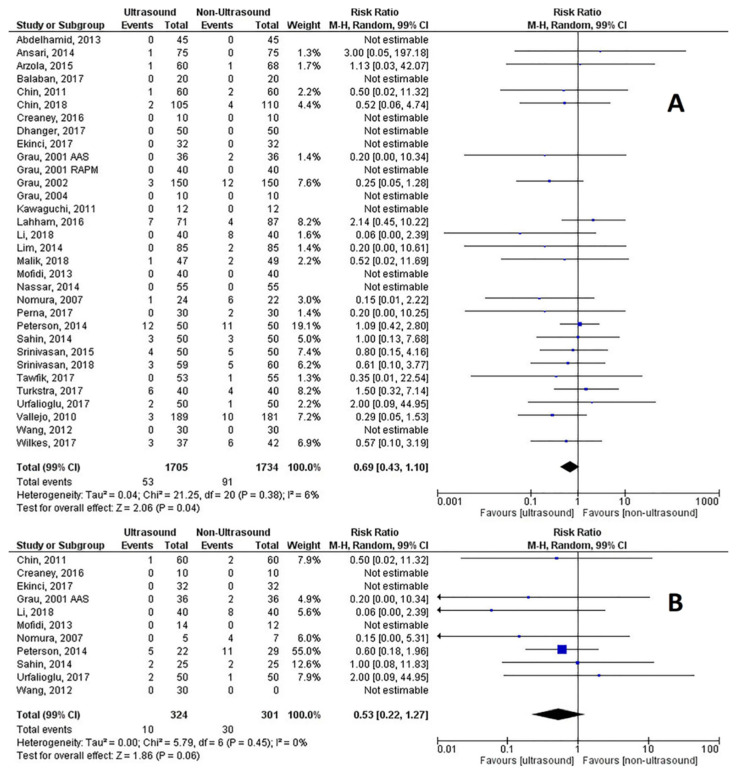
(**A**) Risk of technical failure of neuraxial procedures; (**B**) Risk of technical failure in difficult spines and obese patients.

**Figure 4 healthcare-09-00479-f004:**
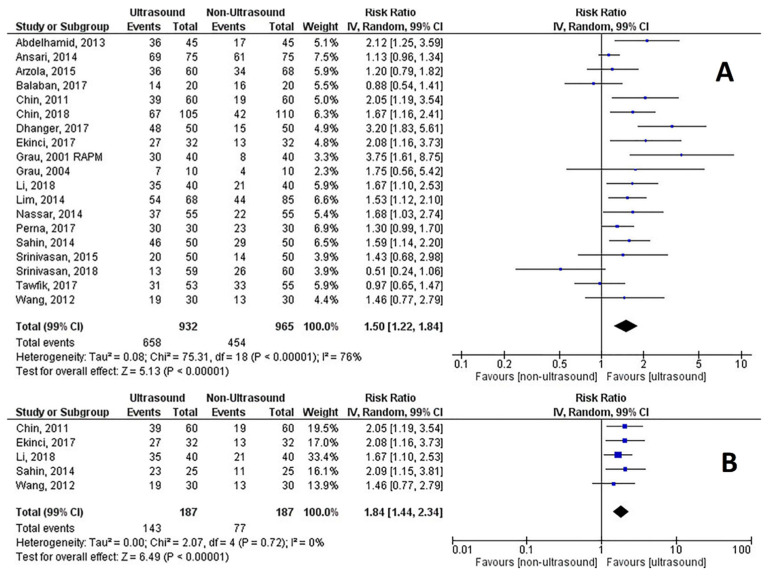
(**A**) First-attempt success rate of neuraxial procedures; (**B**) First-attempt success rate in difficult spines and obese patients.

**Figure 5 healthcare-09-00479-f005:**
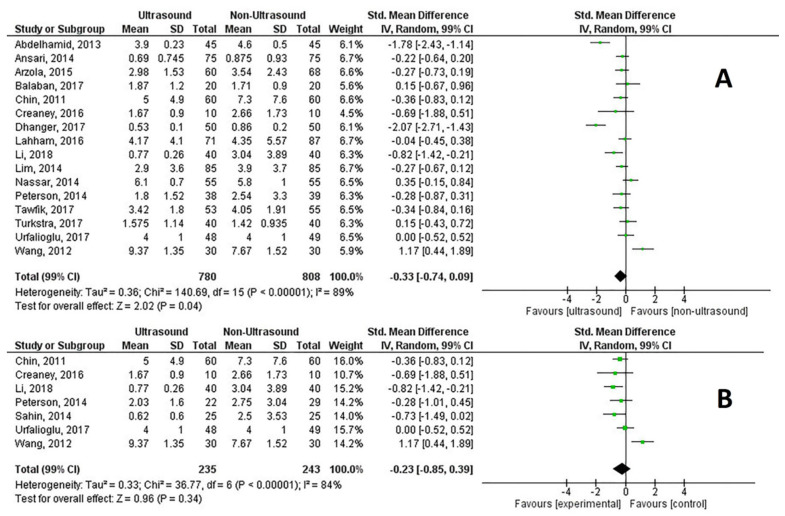
(**A**) Number of needle redirections of neuraxial procedures; (**B**) Number of needle redirections in difficult spines and obese patients.

**Table 1 healthcare-09-00479-t001:** Characteristics of eligible trials.

Author, Year; Country	Patient Population (*n*)	Technique (Methods)	Primary; Secondary Outcomes	Jadad Score
**Abdelhamid, 2013; Egypt [39]**	Adult unspecified	Spinal at L4-5 (US vs. LM group)	Number of needle insertions, number of redirections; procedure time, patient satisfaction	2
**Ansari, 2014; UAE [40]**	Obstetrics CS	Spinal at L3-4 or L4-5 (US vs. LM group)	Spinal procedure time; number of needle insertions/redirections, headache, backache, patient satisfaction	3
**Chin, 2011; Canada [41]**	Orthopedic difficult spine	Spinal (US vs. LM group)	First-attempt success; number of needle insertions/redirections, failure rate, procedure time	3
**Creaney, 2016; Ireland [42]**	Obstetrics, CS, difficult spine	Spinal L3-4 (US vs. LM group)	Number of redirections; total procedure time, needling time, identification time, patient satisfaction	3
**Dhanger, 2017; India [43]**	Obstetrics, CS	Spinal L3-4 (US vs. LM group)	Number of needle insertions; number of redirections, total procedure time, identification time	3
**Ekinci, 2017; Turkey [44]**	Obstetrics, difficult spine	Spinal (US vs. LM group)	Number of needle insertions; number of redirections, number of levels, procedure time, first attempt success, first-pass success	2
**Li, 2018; China [45]**	Obstetrics elective CS, obese	Spinal (US vs. LM group)	First-attempt success; number of insertions/redirections, total procedure time, identification time, needling time, patient satisfaction, number of levels, complications	4
**Lim, 2014; Singapore [46]**	Orthopedic, urologic, general surgery	Spinal (US vs. LM group)	First-attempt success; number of redirections, needling time, paresthesia, traumatic taps, supervisor interventions, patient satisfaction	3
**Sahin, 2014; Turkey [47]**	Obstetrics non-obese and obese	Spinal at L4-5 (US vs. LM group)	First attempt success; number of needle insertions/redirections/levels attempted, failure rate, needling time, paresthesia, headache, backache	3
**Srinivasan, 2015; Ireland [48]**	Orthopedic (THR, TKR)	Spinal (US vs LM group)	Number of redirections; number of attempts, identification time, needling time, level of block, complications, periprocedural pain and discomfort score	3
**Srinivasan, 2018; Ireland [49]**	Orthopedic (THR, TKR)	Spinal (US-guided paramedian spinal L5-S1 vs LM group)	Number of redirections; number of insertions, first attempt success, first-pass success, identification time, needling time, block level at 30′, incidence of radicular pain, paresthesia bloody tap, periprocedural pain or discomfort	2
**Turkstra, 2017; Canada [50]**	Obstetrics CS	Spinal (US vs LM group)	Number of insertions/redirections; procedure time, level of block, need for consultant intervention, paresthesia, bloody tap	4
**Urfalioglu, 2017; Turkey [52]**	Obstetrics CS, obese	Spinal L4-5 (US vs. LM group)	Number of needle insertions, number of redirections; procedure time, needling time, block level, headache, backache	2
**Arzola, 2015; Canada [24]**	Obstetrics labor anlagesia	Epidural (US vs. LM group)	Epidural catheterization time, number of levels attempted, number of needle redirections; total procedural time, first attempt success, number of catheterization attempts, failure rate, patient satisfaction.	3
**Balaban, 2017; Turkey [25]**	Obstetrics labor analgesia	Epidural at L4-5 (US vs. LM group)	Not specified; number of insertions, number of levels attempted, epidural catheterization time, complications	2
**Grau, AAS 2001; Germany [26]**	Obstetrics labor analgesia	Epidural (US vs. LM group)	Not specified; number of redirections, number of levels attempted, catheter advancement attempts, failure rate, headache, backache	2
**Grau, 2002; Germany [27]**	Obstetrics	Epidural (US vs. LM group)	Not specified; number of redirections, number of levels attempted, number of attempts to thread the catheter, VAS scores, rate of incomplete analgesia, headache, backache	2
**Kawaguchi, 2011; Japan [28]**	Orthopedics (THA)	Epidural (US vs. LM group)	Successful ipsilateral dominant epidural block; failure rate, efficacy of ipsilateral-dominant block by PCEA with respect to analgesia and modified Bromage scores	1
**Malik, 2018; USA [29]**	Obstetrics, labor analgesia	Epidural at L5-S1 (US vs. LM group)	Incidence of S2 block 30′ after local anesthetic administration; pain during labor or during delivery, catheter replacement	3
**Perna, 2017; Italy [30]**	Obstetrics labor analgesia	Epidural to L3-4 or L2-3 (US vs. LM group)	Number of attempts (redirections and insertions); number of redirections, number of insertions, first-pass success	4
**Vallejo, 2010; USA [31]**	Obstetrics labor analgesia	Epidural L3-4 or L4-5 (US vs. LM group)	Technical failure rate; number of redirections/reinsertions, unintended dural puncture	3
**Wilkes, 2017; USA [32]**	Obstetrics	Epidural L2-S1 (US vs. US-Sham group)	Pressure pain threshold; number of redirections, number of reinsertions	3
**Chin, 2018; Australia [33]**	Obstetrics, CS	CSE below L1-L2	First-attempt success, procedure difficulty; block quality, patient experience, complications	4
**Grau, RAPM 2001; Germany [34]**	Obstetrics, CS	CSE L3-L4 (US vs. LM group)	Not specified; first attempt success, number of levels attempted, identification time	1
**Grau, 2004; Germany [35]**	Obstetrics	CSE (Real-time US vs. pre-procedural US vs. LM group, 3 groups)	Number of needle insertions; number of redirections, number of levels attempted, incomplete analgesia, complications, patient satisfaction, VAS scores	2
**Nassar, 2014; Egypt [36]**	Obstetrics, Labor analgesia	CSE (US vs. LM group)	First-attempt success; number of insertions, no redirections, procedure time, identification time, needling time	3
**Tawfik, 2017; Egypt [37]**	Obstetrics CS	CSE at L2-3 or L3-4 (US vs. LM group)	First-direction success; first attempt success, number of redirections/ insertions, needling time, patient satisfaction, complications	4
**Wang, 2012; China [38]**	Obstetrics CS, obese	CSE at L3-4 (US vs. LM group)	First-attempt success; procedure time, complications, puncture site hemorrhage	2
**Lahham, 2016; USA [20]**	ER	LP (US vs. LM group)	Number of redirections, number of reinsertions, needling time, failure rate	2
**Mofidi, 2013; Iran [21]**	ER	LP (US vs. LM group)	Procedure time; number of needle insertions, traumatic LPs, pain score	2
**Nomura, 2007; USA [22]**	ER	LP (US vs. LM group)	Success of LP; number of attempts, ease of procedure	4
**Peterson, 2014; USA [23]**	ER	LP (US vs. LM group)	Number of insertions/redirections, success of LP; pain score, needling time, traumatic taps, patient satisfaction	2

Abbreviations and definitions: US: Pre-procedural ultrasound scan; LM: conventional landmark technique (palpation unless otherwise stated); LP: lumbar puncture; ER: emergency room; CS: cesarean section; THA: total hip arthroplasty; TKA: total knee arthroplasty; CSE: combined spinal-epidural anesthesia; first-attempt success: a single needle insertion with or without redirections; identification time: time for identifying landmarks (by US or LM method); procedure time: total procedure time from start to end (palpation/US imaging + needling time) unless otherwise stated. Technical failure rate was defined as the need to (i) use alternative techniques (e.g., use of ultrasound in the conventional landmark group or vice versa) to achieve the effect, (ii) conversion to general anesthesia or (iii) discontinuation of the technique.

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
