# Peer review of "Pre-Procedural Lumbar Neuraxial Ultrasound—A Systematic Review of Randomized Controlled Trials and Meta-Analysis"

_healthcare, 2021, doi:10.3390/healthcare9040479_

Round 1

Reviewer 1 Report

Nice meta-analysis. mostly shows that evidence is low grade or weak for use of preprocedural US before the neuraxial procedure. 

Not sure why 99% CI was used in the analysis.

CI are crossing 1 for failure rate, procedure time and needle time so, US not helpful in improving these outcome. Only first time success rate and redirection improved. 

Does this means US does not improve clinical outcome just patient comfort ? Please state that clearly.

Author Response

We would like to thank the reviewer for taking the time and effort to assess our original submission so meticulously. We have taken into account all of your comments and recommendations and we have modified our paper accordingly. All manuscript changes have been highlighted using the “tracked changes” function provided by Microsoft Word. Detailed replies to the reviewer’s comments are provided below:

Nice meta-analysis. mostly shows that evidence is low grade or weak for use of preprocedural US before the neuraxial procedure. 

Not sure why 99% CI was used in the analysis.

A: We used 99% CI to improve the effect estimation with respect to a 95% CI. We have added a relevant sentence in line 343; it now reads:”… outcomes testing; the latter was implemented to improve the effect estimation with respect to a 95% CI; the…”.

CI are crossing 1 for failure rate, procedure time and needle time so, US not helpful in improving these outcome. Only first time success rate and redirection improved. 

Does this means US does not improve clinical outcome just patient comfort ? Please state that clearly.

A: We thank you for the comment: As we report in the manuscript (page 11, ln 286), we found a 31% reduction in the risk of failure of spinal or epidural anaesthesia in the general cohort and a 47% reduction in the subgroup of difficult spine and obese patients. We deem this as a significant difference worthy of mentioning because of the high failure rate in this subgroup. We are therefore more confident to recommend the use of US in this subgroup (page 11, ln 278) for failure rate. The main risk of increased attempts and traumatic technique to identify the epidural or subarachnoid space are complications, such as spinal hematoma, traumatic cord injury or post-dural puncture headache. We are clearly stating this in lines 289-292. It now reads:” Thus, we recommend pre-procedural ultrasound, to avoid serious complications, associated with increased attempts and traumatic technique to identify the epidural or subarachnoid space, such as spinal hematoma, traumatic cord injury or post-dural puncture headache”.

Reviewer 2 Report

This is a very nice, well written review I just have one major point: 1. A differentiation between “Spinal anaesthesia” and “epidural anaesthesia” would be very interesting and would have more clinical relevance. I would recommend a subanalysis of these to groups.

The use of ultrasound in regional anaesthesia has become more popular in the past couple of years. 

Regional anaesthesia of peripheral nerves as well as of nerve plexus are well established whereas the advantage of ultrasound concerning neuroaxial regional anaesthesia is not so evident and not as widely spread in clinical practice. 

 Arguments named by anaesthesiologic colleagues against the use of ultrasound for neuroaxial regional anaesthesia in routine practice are satisfaction with their success using spinal and epidural anaesthesia in normal landmark technique. 

In the subgroup of „difficult spine and obese patients“ success with the classic landmark technique is certainly reduced. 

Yet, in these difficult patients high expertise using ultrasound is needed to obtain an outcome improvement with lumbar neuraxial regional anaesthesia. 

It is important to point out that the image resolution of the ultrasound devices is constantly improving, therefore being able to visualize difficult anatomic sites better from one generation of the device to the next. 

The review of Sidiropoulou et al. with its promising data shows that it is very rewarding to know this ultrasound technique especially in difficult spine and obese patients. 

Further minor comments: 

The time span of the publications from 1.1.1980 to 1.6.2019 covers 40 years.  

The oldest enclosed publications are from T. Grau from 2001. 

I would request that the authors explain to the reader the difference in the image quality between the ultrasound devices back then and today. Do the authors expect an influence on the former study outcome because of the reduced image resolution? 

Author Response

We would like to thank the reviewer for taking the time and effort to assess our original submission so meticulously. We have taken into account all of your comments and recommendations and we have modified our paper accordingly. All manuscript changes have been highlighted using the “tracked changes” function provided by Microsoft Word. Detailed replies to the reviewers’ comments are provided below:

This is a very nice, well written review I just have one major point: 1. A differentiation between “Spinal anaesthesia” and “epidural anaesthesia” would be very interesting and would have more clinical relevance. I would recommend a subanalysis of these to groups.

A: We would like to thank the reviewer for this very interesting comment. Preprocedural US is used in order to identify relevant anatomy, determined intervertebral space and entry needle point. The end point of the needle (entering the subarachnoid or epidural space) is not identified. Therefore, when using preprocedural US there is no difference if the anesthetic used is spinal or epidural anesthesia. This is why the present review as well as previous reviews (ref 6 and 7) have not differentiated neither these two techniques nor combined spinal-epidural anesthesia or lumbar puncture patients.

It would be completely different if the US group involved “real time US”. During a real time, US procedure, the needle is seen entering the epidural or the subarachnoid space respectively, therefore a differentiation between the two techniques is imperative. We did not include studies involving real time US in this review because the studies published are few and the techniques used are heterogeneous. Hopefully in the future with more RCTs on real time US we will be able to examine the evidence concerning this subject.

The use of ultrasound in regional anaesthesia has become more popular in the past couple of years. 

Regional anaesthesia of peripheral nerves as well as of nerve plexus are well established whereas the advantage of ultrasound concerning neuroaxial regional anaesthesia is not so evident and not as widely spread in clinical practice. 

 Arguments named by anaesthesiologic colleagues against the use of ultrasound for neuroaxial regional anaesthesia in routine practice are satisfaction with their success using spinal and epidural anaesthesia in normal landmark technique. 

In the subgroup of „difficult spine and obese patients“ success with the classic landmark technique is certainly reduced. 

Yet, in these difficult patients high expertise using ultrasound is needed to obtain an outcome improvement with lumbar neuraxial regional anaesthesia. 

It is important to point out that the image resolution of the ultrasound devices is constantly improving, therefore being able to visualize difficult anatomic sites better from one generation of the device to the next. 

A: Thank you for this suggestion. It is true that US has improved in the past 20 years. We have added a comment in the discussion (pg 11, ln 311). It now reads:” Moreover, image resolution of modern US machines is constantly improving, therefore imaging accuracy will hopefully encourage its widespread use.  “.

The review of Sidiropoulou et al. with its promising data shows that it is very rewarding to know this ultrasound technique especially in difficult spine and obese patients. 

Further minor comments: 

The time span of the publications from 1.1.1980 to 1.6.2019 covers 40 years.  

The oldest enclosed publications are from T. Grau from 2001. 

I would request that the authors explain to the reader the difference in the image quality between the ultrasound devices back then and today. Do the authors expect an influence on the former study outcome because of the reduced image resolution? 

A: We have added a comment in the manuscript (see previous answer). The author of all earlier published studies (ref 26, 27, 34 and 35) found significant differences in terms of number of needle redirections and in first attempt success rate. It is expected that, with the advent of newer US machines with better resolution, US imaging of the spine is, consequently, improved. However, an advantage in the use of US was observed even in these earlier studies with their modest imaging. It now reads:” Moreover, image resolution of modern US machines is constantly improving, therefore, imaging accuracy is expected to encourage its widespread use. Interestingly, authors of the earlier published studies [26,27, 34,35], utilizing ultrasound equipment with modest imaging,  also found significant differences in terms of number of needle redirections and in first attempt success rate.“.